# HIERARCHICAL GRAPH-CODING DIFFUSION MODEL WITH ADAPTIVE INFORMATION BOTTLENECK FOR MULTICHANNEL SPEECH ENHANCEMENT

## ABSTRACT

Diffusion models have achieved strong performance in multichannel speech enhancement, especially in unseen noisy scenarios. However, most existing diffusion method rely on globally consistent guidance applied either to the output or uniformly across denoiser layers, which fails to provide layer-specific adaptation and introduces redundancy, thereby constraining denoising performance.To address these challenges,we propose a novel hierarchical graph-coding diffusion model with adaptive information bottleneck (HG-Diff-IB) for multichannel speech enhancement. Specifically, we introduce a hierarchical alignment method to align graph-coding with the denoiser at different depths, together with a layer-wise graph-coding modulation mechanism that injects graph information into intermediate features, enabling layer-specific guidance of diffusion feature distributions. Furthermore, we introduce an adaptive information bottleneck that dynamically adjusts the feature compression according to the estimated SNR, effectively balancing noise suppression and target feature preservation. Experimental results demonstrate that our proposed method outperforms baselines in various evaluation metrics.

## 1 INTRODUCTION

Multichannel speech enhancement has witnessed significant progress in recent years, driven by the synergy of deep learning innovations. By leveraging the spatiotemporal correlations across multiple microphone arrays, it has become a powerful approach for enhancing speech quality in dynamic acoustic environments (Hao et al., 2022; Chau et al., 2024). This advancement not only enables more robust noise suppression, but also delivers better generalization capabilities across diverse real-world scenarios, such as in smart speakers, mobile phones, and hearing aids, where non-stationary noise and interference are prevalent (Doclo et al., 2015; Sainath et al., 2017; Saryuddin Assaqty et al., 2020).

Recently, diffusion models have shown remarkable potential in speech enhancement (Lu et al., 2021; Gonzalez et al., 2024), achieving high-quality reconstruction even under challenging acoustic conditions. By leveraging iterative denoising, these models provide stronger robustness compared to other approaches (Elgiriyewithana & Kodikara, 2024). However, despite these advances, existing diffusion-based SE methods still face two critical limitations.

Firstly, most approaches lack layer-specific guidance, limiting their ability to fully exploit hierarchical representations in denoise process. For example, G-DiffuMSE (Yu et al., 2025) adjust the sampling mean in each denoise step by introducing STGCN reconstruction loss and adversarial loss. NADiffSE (Wang et al., 2023) and FUSE (Yang et al., 2024) introduce condition to guide each layer of the diffusion model, but it provides the same guidance for every layer, resulting in a lack of targeted adaptation.

Secondly, the adopted guidance features often contain substantial redundancy, introducing irrelevant information that may mislead the denoising process. For instance, DOSE(Tai et al., 2023) and CDiffuSE (Lu et al., 2022) directly concatenates noisy speech into the input to guide denoising,

which will inevitably carry redundant noise information and may even mislead the denoiser. DAVSE (Chen et al., 2024) introduce the visual modality as a condition into the audio diffusion, which has a large amount of information and contains redundancy across modalities.

To address the aforementioned challenges, this paper introduces a novel hierarchical graph-coding diffusion model with adaptive information bottleneck (HG-Diff-IB) for multichannel speech enhancement. The main contributions can be summarized as follows:

1) Hierarchical alignment method: aligning shallow and deep graph-coding features with the denoiser encoder and decoder for hierarchical guidance.

2) Layer-wise graph-coding modulation: injecting graph information into intermediate layers of the denoiser for precise feature distribution adjustment.

3)Adaptive information bottleneck: dynamically regulating feature compression according to estimated SNR to balance noise suppression and target preservation.

## 2 METHOD

In this section, we will introduce the proposed hierarchical graph-coding diffusion model, as illustrated in Fig. 1(a). It comprises three main components: 1) hierarchical alignment method, 2) layer-wise graph-coding modulation, and 3) information bottleneck with SNR adaptation for optimization.

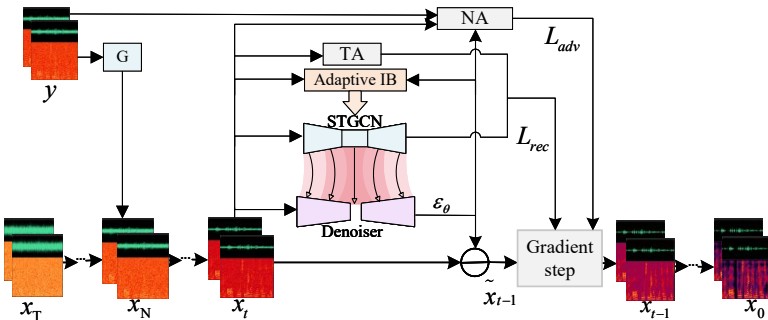

Figure 1: Overview of the proposed HG-Diff-IB framework, including hierarchical graph-coding extraction, layer-wise graph-coding modulation, and adaptive information bottleneck.

### 2.1 HIERARCHICAL ALIGNMENT METHOD

To provide specific guidance throughout the diffusion process, we propose a hierarchical alignment method, aligning hierarchical graph-coding extracted by the STGCN (Hao et al., 2022; Zhang, 2024) with the denoiser's latent features.The denoiser refers to the UNet utilized in the diffusion process.

As shown in Fig.1, shallow graph-coding, containing frame level features and partial phonetic and semantic level features, guides the encoder to extract target speech structures. Deep graph-coding, rich in frame level features, guides the decoder to deliver detailed temporal cues, enabling precise speech reconstruction.

Specifically, we formalize the hierarchical hymmetric alignment as $\mathcal{HA} : \mathcal{I} \longrightarrow J$. Here, $\mathcal{I} = \{1, 2, ..., i, ..., I\}$ denotes the layer index set of hierarchical graph-coding and $\mathcal{J} = \{1, 2, ..., j, ..., J\}$ represent that of the UNet. For any layer index $i \in \mathcal{I}$, its corresponding layer index $j = \mathcal{HA}(i)$ can be written as:

$$\mathcal{HA}(i) = \left\lfloor \frac{J-1}{I-1} \cdot (i-1) + 1 \right\rfloor \tag{1}$$

Through its layer-specific alignment mechanism, this hierarchical alignment enables graph-coding features to be smoothly embedded into the corresponding layers of the denoiser, rather than being crudely fed into a single layer. This integration ensures that at each stage of the entire diffusion process, the denoiser can leverage local detail cues to enhance signal precision and rely on global reconstruction hints to maintain the consistency of speech structure, thereby preventing the generation of distorted speech.

## 2.2 Layer-wise Graph-coding Modulation

Based on the hierarchical alignment method established in Sec.2.1, we further introduce a layer-wise graph-coding modulation mechanism inspired by (Hudson et al., 2024), which directly injects graph-coding information into the intermediate features of the denoiser to achieve adjustment for the denoiser feature distribution.

Specifically, given the $i$-th graph-coding $\mathcal{F}_{\phi,i}(xt)$ and the $\mathcal{HA}(i)$-th layer latent feature of the denoiser $\theta_{\mathcal{HA}(i)}(x_t, t)$, the affine-modulated latent feature is given by:

$$\theta_{\mathcal{HA}(i)}^{'}(x_t, t) = z_s \cdot \text{AdaIN}(\theta_{\mathcal{HA}(i)}(x_t, t)) + z_b \tag{2}$$

Here, $z_s$ and $z_b$ denote the scale and bias parameters derived from a linear projection of $\mathcal{F}_{\phi,i}(x_t)$, which directly modulate the feature distribution. The adaptive instance normalization $\text{AdaIN}(\cdot)$ normalizes each channel independently and is commonly defined as (Huang & Belongie, 2017):

$$\text{AdaIN}(\theta_{\mathcal{HA}(i)}(x_t, t)) = \frac{\theta_{\mathcal{HA}(i)}(x_t, t) - \mu}{\sigma} \tag{3}$$

Here, $\mu$ and $\sigma$ are the mean and standard deviation of the feature $\theta_{\mathcal{HA}(i)}(x_t, t)$, computed independently for each channel.

Speech signals exhibit strong correlation and regularity across multiple channels, while the statistical characteristics of noise are typically more random. AdaIN can normalize the joint features of multiple channels to highlight the consistent patterns of speech features, suppress the random fluctuations of noise, and thereby enable the model to more easily distinguish between speech and noise components.

## 2.3 Information Bottleneck for Optimization

### 2.3.1 Information Bottleneck with SNR Adaptation

To effectively suppress noise while preserving target features, we introduce an adaptive information bottleneck that dynamically regulates the feature compression according to the estimated SNR. It ensures that the model retains sufficient information under high SNR conditions, while enforcing stronger compression to suppress noise under low SNR conditions.

Formally, based on the general form of the information bottleneck loss(Hu et al., 2024), the adaptive information bottleneck loss is defined as:

$$L_{IB} = -I(Z; Y) + \beta_{\text{adapt}} I(Z; X) \tag{4}$$

where $I(\cdot; \cdot)$ denotes mutual information, $\beta_{\text{adapt}}$ controls the tradeoff, large values results in a highly compressed representation, $X$, $Y$ and $Z$ denote the input, output, and latent representation, respectively.

The adaptive parameter $\beta_{\text{adapt}}$ is computed directly from the temporal similarity of the input STFT features. Let $\mathbf{x}_t \in \mathbb{R}^{C \times T}$ be the signal of the $t$-th denoise step, and $W^Q, W^K \in \mathbb{R}^{T \times d_k}$ learned projection matrices. Then $\beta_{\text{adapt}}$ is obtained as

$$\beta_{\text{adapt}} = softmax\left(\frac{W^Q \mathbf{x}_t \cdot W^K \mathbf{x}_t^{\top}}{\sqrt{d_k}}\right) \tag{5}$$

In this way, the adaptive information bottleneck allows the model to flexibly regulate feature compression based on real-time SNR, improving both noise reduction and information preservation in diverse SNR conditions.

### 2.3.2 COLLABORATIVE OPTIMIZATION FOR GRAPH AND DIFFUSION

To prevent potential bias caused by using graph-coding to guide the denoiser alone, we propose a cooperative optimization strategy between the Graph and Diffusion modules. At each denoising step $t$, the intermediate latent feature of the diffusion model is first updated via layer-wise hierarchical graph-coding modulation, according to Eq.2. Then, the graph network is optimized using the loss:

$$L_{IB} = \|\mathcal{F}_\phi(x_t) - x_{0,t}\|_2^2 + \beta_{\text{adapt}}I(Z;X) \tag{6}$$

where $x_{0,t} = \frac{1}{(1-m_t)\sqrt{\alpha_t}}\left(x_t - m_t\sqrt{\alpha_t}y - \sqrt{\delta_t}\epsilon_\theta(x_t)\right)$ is the estimated target at step $t$ by denoiser. Instead of using a single set of graph-coding features to guide all steps, the graph-coding features are dynamically updated with step t. Even if a temporary deviation occurs at a certain step, the graph network optimization process in subsequent steps can quickly correct it, thereby preventing the accumulation of deviations.

## 3 EXPERIMENT

### 3.1 EXPERIMENTAL SETUP

**Datasets:** For dataset construction, we synthesize 6,000 six-channel recordings using the speech from DNS-Challenge Dubey et al. (2022) and noise from ESC50 (Piczak, 2015) dataset via the gpuRIR toolkit (Diaz-Guerra et al., 2021). These recordings simulate general speech-noise scenarios in a 5m×4m×3m room, with a six-channel microphone array placed at the center of the room. For evaluation, we use a test set consisting of 108 samples from DNS-Challenge combined with noise from FSD50K (Fonseca et al., 2021), which includes unseen noise types and covers SNRs ranging from -5 dB to 10 dB.

**Baselines:** Proposed framework are compared with: (1) DM-STGCN-NTA (Zhang, 2024): a framework which employs GCN and spatio-temporal convolution to capture spatial temporal spectral correlations; (2) Diffwave (Kong et al., 2020): a diffusion model with bidirectional dilated conv; (3) DOSE (Tai et al., 2023): a framework which adopts dropout and adaptive prior to address condition collapse; (4) CDiffuSE (Lu et al., 2022): a conditional diffusion which uses the observed noisy speech signal as condition to adapt to non-Gaussian noise; (5) G-DiffuMSE (Yu et al., 2025): a mechanism integrates graph-guided diffusion with noise-conditional modeling for robust multi-channel SE.

**Metrics:** PESQ (Rix et al., 2001)and STOI (Taal et al., 2011) are used to measure speech quality, whose value ranges are [-0.5, 4.5] and [0, 1], respectively.

For pre-training, we initialize Diffusion with VoiceBank (Veaux et al., 2013)-pretrained weights and fine-tune it on synthetic data for 301 epochs with learning rate setted to 1e-4. And we pre-trained DM-STGCN-NTA on synthetic data for 351 epochs with learning rate setted to 1e-4. During the sampling process, we further update the DM-STGCN-NTA using the optimization strategy mentioned in Sec.2.3 for 10 epochs with a learning rate of 1e-6.

### 3.2 RESULTS AND ANALYSIS

Table 1: PESQ results of models with different input SNRs

|  | SNR=-5dB | SNR=0dB | SNR=5dB | SNR=10dB | Avg. |
|---|---|---|---|---|---|
| Noisy | 1.0338±0.0187 | 1.0362±0.0202 | 1.0824±0.0643 | 1.1122±0.0568 | 1.0662±0.0558 |
| DM-STGCN-NTA | 1.0599±0.0347 | 1.1465±0.0930 | 1.2676±0.1320 | 1.4014±0.1523 | 1.2189±0.1708 |
| Diffwave | 1.0331±0.0178 | 1.0443±0.0407 | 1.1104±0.1055 | 1.1461±0.0838 | 1.0835±0.0849 |
| DOSE | 1.0438±0.0197 | 1.0987±0.0548 | 1.1431±0.0587 | 1.2119±0.0605 | 1.0944±0.0754 |
| CDiffuSE | 1.0882±0.1286 | 1.1148±0.1425 | 1.2351±0.1941 | 1.4474±0.1999 | 1.2214±0.2207 |
| G-DiffuMSE | 1.0617±0.0398 | 1.1503±0.0995 | 1.2758±0.1429 | 1.4010±0.1342 | 1.2222±0.1701 |
| HG-Diff-IB(ours) | **1.1088±0.0566** | **1.1971±0.0944** | **1.3171±0.1340** | **1.4357±0.1258** | **1.2647±0.1634** |

### 3.2.1 MULTI-MODEL COMPARISON UNDER DIFFERENT SNRS

Table.1 presents the PESQ performance of all models across different SNR conditions. Overall, our proposed hierarchical graph-coding diffusion model with adaptive information bottleneck consistently outperforms both discriminative and diffusion-based baselines, achieving robust improvements across all noise levels.

As against the discriminative baselines, our method outperforms DM-STGCN-NTA by an average of 3.76%, as well as 4.61% and 4.41% gain at -5dB and 0dB, which verifies its ability to suppress strong noise.

Compared with diffusion-based baselines, it gains average improvements of 16.72%, 15.56%, 3.55% and 3.48% over Diffwave, DOSE, CDiffuSE and G-DiffuMSE, respectively, which shows the robustness of our method in unseen noisy scenarios. Notably, proposed methed shows significant performance gains at -5dB and 0dB: it outperforms Diffwave by 7.32% and 14.64%, DOSE by 6.23% and 8.95%, CDiffuSE by 1.90% and 7.38%, and G-DiffuMSE by 4.43% and 4.07%, confirming robust noise suppression in tough scenarios.

### 3.2.2 ABLATION STUDY

Table.2 and Fig.2 presents an ablation study to assess the contribution of each component in the proposed method.

Table 2: Ablation study on PESQ and STOI

| PESQ | | | | | |
|---|---|---|---|---|---|
| Input SNR | -5dB | 0dB | 5dB | 10dB | Avg. |
| Noisy | 1.0338 | 1.0362 | 1.0824 | 1.1122 | 1.0662 |
| Diffusion | 1.0617 | 1.1503 | 1.2758 | 1.4010 | 1.2222 |
| +FiLM Perez et al. (2017) | 1.0633 | 1.1552 | 1.2861 | 1.4181 | 1.2307 |
| +AdaGN Wu & He (2018) | 1.0608 | 1.1521 | 1.2860 | 1.4314 | 1.2326 |
| +AdaIN Huang & Belongie (2017) | 1.0628 | 1.1579 | 1.2952 | 1.4332 | 1.2373 |
| ++fixedIB | 1.1063 | 1.1963 | 1.3166 | 1.4261 | 1.2613 |
| ++adaptiveIB | **1.1088** | **1.1971** | **1.3171** | **1.4357** | **1.2647** |
| STOI | | | | | |
| Input SNR | -5dB | 0dB | 5dB | 10dB | Avg. |
| Noisy | 0.6539 | 0.7270 | 0.7744 | 0.8385 | 0.7484 |
| Diffusion | 0.6999 | 0.8015 | 0.8507 | 0.8933 | 0.8114 |
| +FiLM | 0.7024 | 0.8006 | 0.8484 | 0.8911 | 0.8106 |
| +AdaGN | 0.6960 | 0.7999 | 0.8490 | 0.8911 | 0.8090 |
| +AdaIN | 0.6972 | 0.8019 | **0.8495** | **0.8923** | 0.8103 |
| ++IB | **0.7308** | 0.8030 | 0.8444 | 0.8811 | 0.8148 |
| ++adaptiveIB | 0.7305 | **0.8069** | 0.8460 | 0.8814 | **0.8162** |

Note: "+" denotes adding a module based on the diffusion module;
"++" is cumulative stacking on top of the previous modules.
FiLM is a general-purpose conditioning method for neural networks that performs feature-wise affine transformations on the network's features based on conditioning information.
AdaGN is an innovative normalization method that integrates time-step and class embedding information into the group normalization operation.

It is obvious that incorporating layer-wise graph-coding modulation performs better than the diffusion baseline on both metrics. For PESQ, it achieves improvements of 0.70% with FiLM, 0.85% with AdaGN, and 1.24% with AdaIN over original diffusion. In terms of STOI, there are enhancements of 1.49% with FiLM, 1.32% with AdaGN, and 1.15% with AdaIN. These results verify the effectiveness of hierarchical graph-coding modulation and also highlight the superiority of AdaIN.

Furthermore, adding the information bottleneck module further enhances both PESQ and STOI results. For PESQ, improvements are observed particularly under extremely noisy environments, for inatance, 4.09% at -5dB and 3.32% at 0dB, verifying its ability to suppress redundant noise information. In terms of STOI, there are also gains, such as 4.43% at -5dB and 2.05% at 0dB.

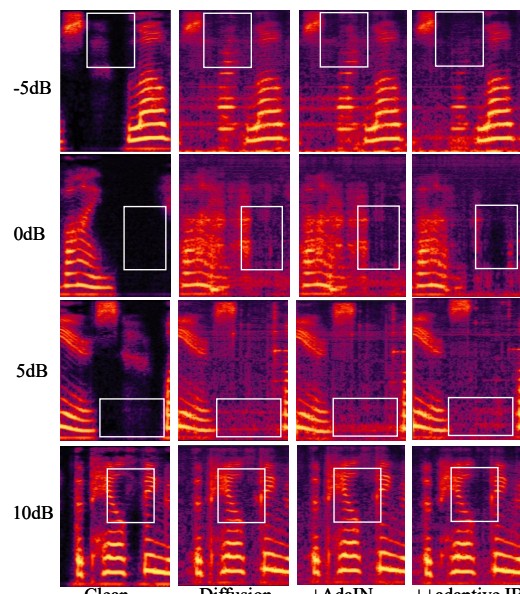

Figure 2: Spectrogram Comparison in Ablation Study

While a fixed information bottleneck causes minor degradation in PESQ and STOI at high SNR, the adaptive IB eliminates this drawback and maintain stable performance on average SNR levels. It can also be observed from the spectrogram shown in Fig.2 that the adaptive IB module can achieve background noise suppression and gain much sparser backgrounds under all SNR conditions.

## 4 CONCLUSIONS

This paper presented HG-Diff-IB, a hierarchical graph-coding diffusion model with adaptive information bottleneck for multichannel speech enhancement. The proposed method enables precise feature modulation through layer-wise graphcoding and adaptively balances noise suppression with feature preservation. Experimental results show improvements over both discriminative and diffusion-based baselines, especially under challenging scenarios.

ACKNOWLEDGMENTS

The work is supported by National Natural Science Foundation of China(Grant No. 62271039). We appreciate the constructive comments from the anonymous reviewers, which help improve this work.

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
