# OpenReview forum: "Hierarchical Graph-coding Diffusion Model with Adaptive Information Bottleneck for Multichannel Speech Enhancement"
_ICLR.cc/2026/Conference — Submitted to ICLR 2026_

### Official Review · Reviewer_TwDz · 2025-10-27

**Soundness:** 3
**Presentation:** 3
**Contribution:** 2
**Rating:** 4
**Confidence:** 4

**Summary:**

This work proposes HG-Diff-IB, a hierarchical graph-coding diffusion model with an adaptive information bottleneck for multichannel speech enhancement. The method introduces hierarchical alignment and layer-wise graph-coding modulation to provide layer-specific guidance for diffusion features, and employs an adaptive information bottleneck that adjusts feature compression based on estimated SNR. Experiments show that HG-Diff-IB achieves superior performance over baseline methods across multiple evaluation metrics.

**Strengths:**

- **Interesting Topic.** The paper focuses on a timely and increasingly popular research topic. In recent years, many researchers have explored integrating generative models (such as diffusion models) with traditional regression-based tasks like speech enhancement, making this work relevant to ongoing trends in the field.
- **Performance.** The proposed model consistently outperforms many advanced baselines across multiple evaluation metrics.

**Weaknesses:**

- **Unclear Motivation for Graph-Coding.** The motivation for introducing graph-coding is not clearly explained. It remains unclear why a GNN is needed for the SE task. The authors do not provide sufficient justification for this design choice, nor do they include implementation details that would help readers understand how the graph structure is constructed or how it benefits the model. As a result, the role and necessity of the graph-coding module remain ambiguous.

- **Missing Definitions and Clarity.**  Several symbols in the paper are undefined or insufficiently explained. For example, in Equation (6), it is unclear what $m_t$ and $y$ represent, and why a part of $y$ is subtracted. Moreover, the formulations of the forward and reverse diffusion processes are not clearly described. These missing definitions and unclear explanations make it difficult for readers to fully understand the methodology and reproduce the results. I suggest the authors carefully review all mathematical expressions and ensure that every symbol is explicitly defined in the main text.

- **Lack of Efficiency Analysis.** The paper focuses primarily on performance improvements over baselines but lacks any analysis or discussion of efficiency. Important aspects such as model complexity (e.g., parameter count, FLOPs) and inference time are not reported. Without these comparisons, it is difficult to evaluate whether the observed performance gains are achieved through better modeling design or simply by increasing computational cost. An explicit efficiency analysis would make the contribution more convincing and practically relevant.

**Questions:**

- **Eq. 5. (Lines 159-161)** The authors state that $\beta_{\text{adapt}}$ allows the model to regulate feature compression based on real-time SNR. However, if the goal is to model the relationship with SNR, why do the authors choose to use $x_t$ rather than $t$ (or an explicit SNR estimate)? Since t already has a monotonic relationship with the SNR in diffusion-based formulations, conditioning on $t$ would seem more straightforward and interpretable. Could the authors clarify this design choice?

---

### Official Review · Reviewer_eHQd · 2025-10-28

**Soundness:** 2
**Presentation:** 2
**Contribution:** 2
**Rating:** 2
**Confidence:** 4

**Summary:**

This paper proposes a hierarchical graph-coding diffusion model with adaptive information bottleneck for multichannel speech enhancement, and shows the effectiveness with experiments on synthesized datasets. The proposed method demonstrates superior performance on two metrics to measure speech quality.

**Strengths:**

- Reasonable proposal of the new method for speech enhancement based on diffusion model.

- Experiments support the proposed method with relative superiority to the comparing methods.

**Weaknesses:**

- The proposed method requires a thorough (maybe theoretical) justification.

- The experiments are too limited to draw any interesting conclusions. Larger and more datasets are required to validate the proposed method.

**Questions:**

- What is the main contribution of the proposed method? Simple adoption of hierarchy would not be enough to justify the novelty of the proposed method.

- How can you guarantee the superiority with the larger benchmark datasets? Simple synthesis of a new dataset would limit the verification of the experiments. More metrics should be introduced to provide the in-depth analysis of the results for deeper discussion.

---

### Official Review · Reviewer_H7Q7 · 2025-10-31

**Soundness:** 1
**Presentation:** 2
**Contribution:** 2
**Rating:** 0
**Confidence:** 5

**Summary:**

Summary: this paper proposes a hierarchical graph-coding diffusion model with adaptive information bottleneck for multi-channel speech enhancement. Contributions are two-fold. First, they propose a hierarchical alignment method to align gradh-coding with the denoiser at different depths. Besides, they introduce an adaptive infomration bottleneck that can adaptively adjust the feature compression at different timesteps.

**Strengths:**

Strengths:
1. They propose a hierarchical graph-coding method to align shollow and deep graph-coding features with the denoiser encoder and decoder.
2. They propose a layer-wise graph coding modulation to inject graph information into intermediate layers.
3. They propose an adaptive information bottleneck to regulate feature compression at different timesteps.

**Weaknesses:**

Weakness:
1. The authors show rather negative attitude on the presentation of the paper. For example, in Fig. 1, no clear illustration is given toward the training/inference pipeline, and no illustration is provided toward the abbreviations listed in the figure, making the readers quite confusing. Besides, no related works, and no preliminary demonstrations toward the problem formulation and existing challenges are provided. I think such a sumit is a pure waste of time and should be desk-rejected.
2. I have trained all the baselines the authors listed in Table 1, and the reported PESQ scores seem anomalous, especially considering the spatial feature can be utilized.
3. Table 2, the improvements brought by IB and adaptiveIB seem marginal and inadequate to validate the claim.

**Questions:**

Questions:

1. Please rewrite the whole paper, and check your code to ensure all the results you reported are valid.
2. Practice your writing skill in research paper and do not throw a meaningless manuscript in ICLR.

---

### Official Review · Reviewer_PFfg · 2025-11-01

**Soundness:** 2
**Presentation:** 1
**Contribution:** 2
**Rating:** 2
**Confidence:** 4

**Summary:**

The paper proposes HG-Diff-IB, a multichannel speech enhancement (SE) framework that combines (i) hierarchical alignment between a graph-coding network (STGCN) and a diffusion UNet, (ii) layer-wise feature modulation using AdaIN with scale/bias predicted from the graph features, and (iii) an adaptive information bottleneck (IB) whose trade-off parameter varies with an SNR proxy derived from self-similarity of STFT features. Experiments on synthetic 6-mic arrays show modest gains in PESQ and small changes in STOI over diffusion and graph/diffusion baselines.

**Strengths:**

1. The proposed approach is simple yet interesting, combining well-established mechanisms (hierarchical conditioning, AdaIN modulation, and information bottleneck)
2. The paper includes a comprehensive evaluation with ablation study that assess the contribution of each module.

**Weaknesses:**

1. The contribution appears incremental relative to existing work. Similar ideas have been explored in diffusion models guided by external features across layers.
2. The paper is poorly written with multiple grammatical errors and typos (e.g. “methed”, “setted”). More importantly, several architectural and training details are missing (UNet and STGCN configuration, STFT parameters, hyperparameters, optimizer, batch size etc).
3. The dataset used is fully synthetic, which limits the conclusions. Also, the metrics used (PESQ/STOI) are not sufficient for perceptual evaluation - a mean opinion score would be better.
4. The performance gains are small, and at high SNR the improvements seem to vanish.

**Questions:**

1. How is I(Z;X) computed in Eq. (6)? e.g. MINE, InfoNCE.
2. Were the baseline models fine-tuned on the same synthetic dataset before comparison? If not, how was fairness ensured?
3. Can you show results on real-world data to show the generalization abilities of your model?
4. How does the model’s performance scale with different microphone counts (e.g. 2, 4, or 8 mics)? Also, how sensitive is the model to the \beta_{adapt} value?
5. Have you seen which noise types the model performs best/worst? Can you show these results?

---

### Meta-Review · Area_Chair_KUsg · 2026-01-06

**Summary:**

Across reviews, the key concerns motivating a negative recommendation were that the paper's technical contribution is perceived as incremental without a compelling novelty/justification story for why graph-coding is necessary, and that the manuscript quality and clarity are substantially below the bar (confusing/underspecified pipeline and symbols, unclear diffusion/IB formulations, and missing reproducibility-critical details such as model/STFT/training configuration). Reviewers also questioned the empirical credibility and scope, which reviewers felt is insufficient for perceptual claims and limits conclusions about real-world generalization.

**Reviewer Concerns:**

All the reviewers have raised the concerns on the quality of the manuscript in its approach, presentation, organization, and etc. I see that the manuscript is far from being ready to be published.

**Reviewer Scores:**

N/A

---

### Decision · Program_Chairs · 2026-01-26

Reject